# Contributions of Letter-Speech Sound Learning and Visual Print Tuning to Reading Improvement: Evidence from Brain Potential and Dyslexia Training Studies

**DOI:** 10.3390/brainsci7010010

**Published:** 2017-01-18

**Authors:** Gorka Fraga González, Gojko Žarić, Jurgen Tijms, Milene Bonte, Maurits W. van der Molen

**Affiliations:** 1Department of Developmental Psychology, University of Amsterdam, Amsterdam 1018 WS, The Netherlands; j.tijms@uva.nl (J.T.); M.W.vanderMolen@uva.nl (M.W.v.d.M.); 2Rudolf Berlin Center, Amsterdam 1018 WS, The Netherlands; 3Department of Cognitive Neuroscience, Maastricht University, Maastricht 6200 MD, The Netherlands; gojko.zaric@maastrichtuniversity.nl (G.Ž.); m.bonte@maastrichtuniversity.nl (M.B.); 4Maastricht Brain Imaging Center, Maastricht University, Maastricht 6200 MD, The Netherlands; 5IWAL Institute, Amsterdam, Amsterdam 1001 EW, The Netherlands; 6Amsterdam Brain and Cognition, University of Amsterdam, Amsterdam 1018 WT, The Netherlands

**Keywords:** reading, dyslexia, ERP, MMN, N170, letter-speech sound integration, associative learning

## Abstract

We use a neurocognitive perspective to discuss the contribution of learning letter-speech sound (L-SS) associations and visual specialization in the initial phases of reading in dyslexic children. We review findings from associative learning studies on related cognitive skills important for establishing and consolidating L-SS associations. Then we review brain potential studies, including our own, that yielded two markers associated with reading fluency. Here we show that the marker related to visual specialization (N170) predicts word and pseudoword reading fluency in children who received additional practice in the processing of morphological word structure. Conversely, L-SS integration (indexed by mismatch negativity (MMN)) may only remain important when direct orthography to semantic conversion is not possible, such as in pseudoword reading. In addition, the correlation between these two markers supports the notion that multisensory integration facilitates visual specialization. Finally, we review the role of implicit learning and executive functions in audiovisual learning in dyslexia. Implications for remedial research are discussed and suggestions for future studies are presented.

## 1. Introduction

Alphabetic written languages consist of symbols of arbitrary form or letters, which are associated by social agreement to specific speech sounds, i.e., perceptually distinct units of spoken language that differentiate between the words (e.g., *b*at, *c*at, *h*at). A broader definition of alphabetic scripts includes abjad scripts (e.g., Hebrew) in which commonly only the consonants are written, syllabic scripts (Japanese katakana) in which single characters represent syllables, and abugida scripts (e.g., Ethiopic Ge’ez) in which vowels do not have their own symbol but are represented by changes in the symbols of the consonants. Languages using alphabetic scripts can vary in the consistency of the connection between grapheme and phoneme from highly transparent (shallow) orthographies (e.g., Finnish, Italian) with high regularity in the letter-speech sound connections [1], to opaque (deep) orthographies (e.g., English, Danish) in which the letter-speech sound correspondences are complex and ambiguous [2,3]. Thus, the first task of a beginning reader in an alphabetic orthography is to match the distinctive visual symbols (graphemes) to the units of speech sound (phonemes). Efficiently establishing letter-speech sound (L-SS) associations, together with the ability to distinguish separate phonemes, so called phonemic awareness, is essential to connect the spelling of written words to their pronunciation and meaning. This ultimately enables sight word reading, that is, automatic and accurate word reading from memory [4]. The majority of the population easily attains the cognitive steps needed for fluent reading with reading instruction and training. However, a small, but significant, percentage of the population diagnosed with developmental dyslexia fails to develop typical levels of reading abilities in the absence of other cognitive or neurological impairments [5]. Developmental dyslexia affects around 5%–10% of children and is characterized by dysfluent and inaccurate word recognition, spelling, and phonological decoding [5,6,7]. The lack of reading fluency is its most persistent symptom. It is hardly remediated by current treatments [8,9,10], although there are inter-individual differences in the level of reading (dys)fluency among dyslexic readers [11,12]. The present review is concerned with impairments in L-SS learning and visual specialization during the initial phases of reading in dyslexic children. Before these deficits are considered, let us first introduce in more detail the key cognitive processes involved in the typical development of fluent reading.

We consider the acquisition of fluent reading skills as a multisensory problem in which a novice reader has to combine a visual element (letter) with an auditory element (speech sound) into a new audiovisual unit. Rather than mapping orthography directly to existing phonological representations of speech sounds, there is an ongoing recoding of the relevant spoken language elements, as it is unlikely that the speech sounds have isolated representations before learning to read [13,14,15]. At the same time, the reader must also learn how to extract a specific speech sound from the otherwise continuous auditory input to create a “graphoneme”, i.e., a new representation of the speech sound that also includes its orthographic characteristic [16]. Ultimately, instead of exclusively connecting the orthographic forms to phonology, expert readers also connect orthographic forms to meaning [17,18]. This facilitates the reading of familiar words, especially if they contain irregular L-SS correspondences [19]. Thus, concurrently with the development of L-SS integration, cognitive changes related to processing print and whole words occur.

There are several lines of evidence showing that experienced readers perceive a visually presented word in a different manner than merely processing its parts separately in succession. For example, the time needed to recognize a word is not always influenced by its length [20,21] and recognition is almost perfect, even when a word is presented for a very brief period of time, such as 50 ms [22]. In addition, a letter is recognized faster if it is embedded in a word than in isolation or as part of a random letter string; this is called word superiority effect [23]. This effect extends to pseudowords, which suggests that it depends partly on prelexical processing and it can be easily generalized to novel stimuli [24]. Moreover, these effects are invariant to differences in size, case, and font, and are influenced by priming across such variations [24]. Importantly, the above findings indicate that there is a shift from primarily relying on phonological recoding strategies to predominantly sight word reading, depending on the familiarity of the word [23,24,25,26,27]. While familiar words are read without involvement of phonology, unfamiliar words and pseudowords would require orthographic-phonological decoding (e.g., [28,29]). Further, these cognitive strategies may depend on the level of orthographic transparency, with beginning readers adopting strategies based on L-SS conversion in more transparent orthographies, while relying more on direct visual recognition strategies in opaque orthographies [30].

The most prevalent view of dyslexia is the phonological deficit theory, based on the finding that dyslexic readers usually exhibit impaired phonological awareness and limited verbal working memory [31]. Within that framework, it is still a matter of debate whether they cannot build clear phonological representations or cannot efficiently use otherwise intact phonological representations [32]. Despite being a widely accepted theory, a purely phonological account of dyslexia does not explain some of the empirical results available. For example, an L-SS deficit is found to be present in the absence of obvious impairments in phonological awareness [33]. That study also showed that children who would later become impaired readers often did not show phonological awareness problems before the onset of reading instruction in kindergarten. Moreover, studies on integration of L-SS correspondences have shown impairments in dyslexics in behavioral [18,33,34], as well as in neural measures [35,36,37,38,39,40,41]. Additionally, a recent study in pre-reading children, at varying risks of dyslexia, revealed a relation between neural L-SS integration and the learning rate/achievement on an artificial L-SS correspondences training [42]. It should also be noted that interpreting the results from phonological tasks as reflecting purely phonological processing is not as straightforward as it may seem. Several studies have shown that orthographic influences can be observed in verbal phonological tasks [43,44,45,46]. This observation suggests that, even without presenting written stimuli, orthographic knowledge influences phonological judgments [47]. This point does not necessarily challenge the phonological deficit idea *per se*, but it stresses the interaction between written and spoken language representation with reading development. Based on the above and similar studies, some authors proposed a failure to successfully develop fluent and automated L-SS associations as a most proximal cause for dyslexia [27,47,48,49,50]. Accordingly, it is suggested that, although dyslexic children do possess letter knowledge already at the end of the first grade, their L-SS integration may not automatize, leading to slower and dysfluent reading [49]. This proposition does not exclude, however, the possibility of phonological deficit in dyslexic readers, but considers it as a more distal cause of reading failure [33] whose influence may be greater in opaque than in transparent languages [51].

From a neurobiological perspective, literacy is subserved by two evolutionary developed neural systems, predominantly located in the left hemisphere [52]: one for spoken language [53,54], and the other for visual object recognition [55]. Since reading is a recent cultural invention, these systems have to adapt to the newly-obtained reading skill [56,57,58]. More specifically, studies have delineated the networks of brain regions involved in L-SS integration and visual word recognition. Brain regions in the superior temporal cortex are involved in multisensory L-SS integration [49,59,60] while a ventral left occipito-temporal region, more specifically the left fusiform gyrus, is involved in fast visual word recognition [24,61]. Additionally, learning to read seems to enhance brain activation in a widespread network [57,58] including regions involved in phonological processing [59,62,63]. The proposed developmental trajectory of brain posterior systems for reading suggests an early specialization of multisensory integration regions that facilitates specialization of the visual occipito-temporal system for word recognition [64]. It is these two posterior brain systems that neuroimaging studies converge on reporting atypical activation in both dyslexic adults and children [9,37,65,66,67]. The findings of those studies are interpreted to indicate deficits in multisensory integration and fast visual word recognition. The relative contribution of each these two systems to reading deficits in dyslexia is addressed in our joint analyses of electrophysiological responses associated with L-SS integration and visual specialization (see Section 2).

In the first part of this paper we review and extend the work of our group regarding reading fluency and its improvements associated with training. The focus is predominantly on electrophysiological measures of L-SS integration and visual word recognition. We start by reviewing our previous findings concerned with a comparison between dyslexics and typically reading children [38,68]. Then, we review our longitudinal findings pertaining to the predictive power of our measures in relation to improvements associated with the first stages of a treatment program. The major focus here is on L-SS associations [40,69,70]. We will then review behavioral data obtained when children received extended practice enhancing their knowledge at the word level [71]. This is followed by examining the relation between electrophysiological measures of audiovisual integration and visual specialization. In the second part of our review, we consider the role of associative learning in establishing L-SS associations within the context of current training approaches. We present a concise review of behavioral and neuroimaging studies of dyslexia on implicit learning and executive functions that we consider relevant to associative learning. We discuss how neurocognitive measures derived from those studies may help to improve the learning of L-SS associations in dyslexics and can be used to predict treatment outcome. Finally, we will present our ongoing EEG research on the learning of artificial L-SS associations.

## 2. Electrophysiological Markers for Letter-Speech Sound Integration and Visual Specialization in Dyslexic Readers

Electroencephalography (EEG) has a high temporal resolution, which makes it a powerful tool to investigate the temporal dynamics of neural activity during fluent reading. Previously, we employed a classical event-related potential (ERP) analysis [38,68] and, more recently, connectivity measures derived from EEG oscillations [72,73], to examine differences between dyslexics and typical readers. In our ERP studies, we used two experimental paradigms to investigate multisensory integration and visual specialization for words, i.e., a crossmodal oddball paradigm [38,40] and a visual word recognition paradigm [68,69], respectively (see Figure 1). These two paradigms should allow a comprehensive picture of the differences between typical readers and dyslexic children and how these differences scale with the severity of the reading (dys)fluency.

The crossmodal oddball paradigm was used to assess mismatch negativity (MMN) and late negativity (LN) responses in relation to developmental dyslexia. The MMN is a fronto-central negativity elicited at around 100–200 ms by an infrequent sound (deviant, or oddball), that is presented within a sequence of a frequent, repeated (standard) sounds [74]. The LN is a later negativity within a broader time range that is usually found around 300–700 ms after the onset of the deviant stimulus. We used a crossmodal version of the oddball paradigm, which involves an auditory-only (phoneme presentation) and an audiovisual condition (phoneme and grapheme presentation), as presented in Figure 1A. In the audiovisual condition, letter and speech sounds were presented either synchronously or the letter preceded the speech sound by 200 ms. The MMN response is believed to represent automatic change detection that is sensitive to deviation from established traces in auditory short-term memory [74,75]. The LN is hypothesized to reflect the more cognitive and/or attentional aspects of this integration, although its functional role is still unclear [40]. Previous studies showed that the simultaneous presentation of letter and speech sounds evoked an MMN but not an LN in adults, whereas it affected the LN in the beginning readers, and both the LN and the MMN in advanced school-aged children [76,77]. This developmental dissociation suggests that, even in a fairly transparent language like Dutch, it may take years to develop the integration of accurate letter-speech sound identification and discrimination at the neural level [38,76,77] while, at the behavioral level these associations are typically observed after only one year of reading instruction [78].

We used a visual word recognition paradigm to examine visual specialization for print. The paradigm consists of a one-back task in which words and strings of symbols are visually presented (see Figure 1B). The ERP component of interest in this paradigm is the visual N1/N170 (hereafter N170), an occipito-temporal component of negative polarity and peak latencies around 200 ms after stimuli. Previous studies found enhanced N170 amplitudes to orthographic compared to visually matched stimuli [79,80]. In a series of studies, Maurer and colleagues examined N170 responses to relative words vs. strings of icon-like symbols in dyslexics and typical readers at different stages of reading acquisition [81,82]. Typical readers showed a left-lateralized N170 amplitude to words vs. symbol strings that increased from kindergarten to 2nd grade, but leveled off between 2nd grade and 5th grade [82]. This was interpreted to suggest an inverted “U” model of development of perceptual visual expertise for print. In the same study, dyslexics showed reduced word-specific N170 amplitudes as compared to typical readers in 2nd grade, while in 5th grade this group difference was not significant, and even an opposite trend was observed at that stage [82]. This finding was interpreted to suggest a protracted developmental trajectory of visual expertise for words in dyslexics. However, other studies using a similar paradigm suggested that visual specialization deficits may remain in dyslexic pre-adolescents [83] and adults [84,85]. Finally, correlations between N170 amplitudes and reading abilities were reported in typical readers [86], and in a group including both dyslexic and typical readers [87,88].

### 2.1. Crossmodal MMN and Visual N170 Related to Reading Fluency in Dyslexic Children

In our study of L-SS integration in dyslexic and typically reading children [38], we used the same crossmodal paradigm (Figure 1A) as in the previous studies [35,76]. The goal of our study was to extend previous findings to an age group between complete beginners and more experienced children, i.e., nine-year-old typical and dyslexic children [38]. The focus of this study was on individual differences in reading fluency, thus, the dyslexic sample was divided into two groups based on their reading fluency score: severely dysfluent dyslexic children and moderately dysfluent dyslexic children. First, we found both MMN and LN effects in the typically reading children, which are consistent with the notion of a slow/protracted development and a high sensitivity to reading related stimuli (e.g., phonemes [89] and words [82,87]) in the first years of reading acquisition; Second, our results revealed a different pattern of integration deficiency depending on the level of reading fluency impairment. We observed a deficit in crossmodal MMN responses that was only present in severely dysfluent dyslexics, while the later crossmodal LN response was reduced in both dyslexic groups compared to typical readers. Importantly, reading fluency scores were correlated with the latency of the crossmodal MMN response when letters and speech sounds were presented simultaneously [38]. The results were interpreted in the framework of an unsuccessful representation of an audiovisual stimulus in severe dyslexics and unavailability of the represented stimulus for further manipulation in both dyslexic groups [37,90,91]. Further, the findings on crossmodal LN might point at deficits in cognitive, explicit associative, and/or attentional processes associated with L-SS integration, rather than an isolated pure bottom-up deficit [35,76,92,93]. This is supported by two recent ERP studies on L-SS integration in dyslexic children reporting abnormal responses suggestive of greater cognitive effort rather than automaticity [94] and a delay in using orthographic and phonological information during response selection [41].

In a companion study we examined electrophysiological responses in typical and dyslexic readers during a visual word recognition task [68]. We presented words vs. strings of meaningless letter-like symbols (Figure 1B) to examine the specificity of the N170 component in nine-year old children. Both typical and dyslexic readers showed a more pronounced N170 component for words compared to symbols. Interestingly, the N170 responses to words were reduced at the left occipito-temporal sites relative to the right hemisphere in typical readers but there was no hemisphere effect in the dyslexic group. We interpreted this finding to suggest facilitated lexical access or more efficient attention allocation in the group of typical readers. Interestingly, N170 amplitudes and reading fluency were positively correlated in the dyslexic group. We concluded that these results point to the functional role of N170 in visual word specialization [79,80] and its potential use to discriminate between typically reading and dyslexic children [82,95].

### 2.2. Crossmodal MMN and Visual N170 as Predictors of Treatment Response

The next question we addressed was whether the training we previously used to enhance reading fluency in dyslexic children [70] would also affect the brain potential responses that we observed to be associated with reading fluency. The training approach was inspired by the multisensory integration deficit account of dyslexia, proposing a deficit in automation of L-SS associations [49]. Most treatments for dyslexia have a strong focus on accurate learning of L-SS correspondences [96,97,98]. However, as previously indicated, this may constitute only a first step towards the automation that L-SS mappings require to be efficiently used for fluent reading [99]. Thus, traditional interventions may not account for the time demands of multisensory integration at the neural level, which seems to take place within a very brief time window in skilled readers [76,77]. Accordingly, we employed a remediation procedure that addresses these automation demands by massive and intentional repetitive training of L-SS correspondences [40,69,70]. The training in automaticity was combined with explicit instruction to establish a strong understanding of phonemic and orthographic regularities as well as decoding skills. These two treatment elements are proposed to enable dyslexics to take advantage of increasing reading experience, facilitating the neural tuning processes required for fluent word identification. Our previous randomized controlled trial study showed that, first, the training led to reading fluency gains; second, L-SS knowledge was not associated with these gains; and, thirdly, initial mapping fluency strongly limited the acquisition of reading speed in untrained dyslexics but not in trained dyslexics [70].

Our ERP training studies investigated whether nine-year old dyslexic children would exhibit changes in crossmodal MMN and LN responses and occipito-temporal N170 following the systematic training of L-SS integration, and whether these changes would relate to reading improvements [40,69]. Regarding crossmodal MMN, we observed moderate improvements in the neural integration of letters and speech sounds after training, particularly in the timing of the late response (LN window) [40]. The early integration response (MMN window) seems to be more treatment resistant as no changes were found after training. Interestingly, however, individual differences in crossmodal MMN timing prior to training related to reading fluency both at pretest and posttest, and predicted gains in reading fluency. In other words, children who exhibited a longer lasting MMN response at pretest read more fluently already before the start of the training and remained more fluent at the second measurement. Importantly, these children also benefitted more from the L-SS training [40]. In relation to visual specialization, we asked whether training would result in changes in the N170 response to words [69]. Interestingly, children that significantly improved in reading fluency showed a significant reduction in visual N170 amplitudes for words relative to the pretest, while this effect was absent in children that did not improve. Furthermore, we found a moderately positive relation between gains in reading fluency and the training-related reduction in N170 responses.

## 3. Current Results: Additional Analysis on MMN and N170

### 3.1. Prediction of Reading Gains after Extended Practice

The previous training was largely based on the initial stages of a longer remedial program implemented at the IWAL institute focusing on L-SS associations [71]. The subsequent phases of the program advanced towards more syllabic and word knowledge, and learning of morphological elements that are productive for reading and spelling of polysyllabic words in Dutch (for more details of the subsequent modules of the program see [10]). From the dyslexic sample of our previous ERP training studies, 15 participants continued with the full program. The mean (SD) age of this subset of participants was 9.07 (0.45) years old. They followed an average (SD; range) of 36 (6; 26–44) additional sessions with an intensity of one session per week. Thus, in total, participants completed 70 (6; 60–78) sessions including the initial letter-speech sound training. Participants did not finish the program at the fixed pace (same number of sessions) as the program goal is to achieve a mastery level for each element of the program (i.e., around 80% of correct answers; [71]). For the present review, we considered data pertaining to reading fluency after the completion of the program. Reading fluency was measured by word (high and low frequency) and pseudoword reading subtests of the 3DM, a standard Dutch battery of reading related tests [78]. Figure 2 shows gains in reading fluency in normative scores after the initial training on L-SS associations (T2) and moderate gains (not significant) after completion of the full reading remediation program (T3). The reading scores for each test and time point and their comparisons are included as supplementary material (see Table 1). The improvement in reading fluency was substantial after the experimental program (T1 to T2: *t* (14) = −4.42, *p* = 0.001; see [70] for the complete assessment). The second part of the program, with the focus on morphology, was followed by moderate improvements in word reading fluency (high and low frequency words combined, T2 to T3: *t* (14) = −2.35, *p* = 0.017), but not for pseudowords (T2 to T3: *t* (14) = −0.66, *p* = 0.260). The more marked improvements after the first part of the program are in line with the specific goals of each of the training parts [100], and with our expectations that the first part would be crucial to break through the initial ‘fluency barrier’ and facilitate further improvements on the trained material, i.e., words rather than pseudowords.

The next step was to examine the reading gains from the beginning until the completion of the remedial program (T1 to T3) using our two ERP markers. We used standardized scores of reading fluency (T scores; M = 50, SD = 10) to account for potential differences in age and provide with a more clinically meaningful estimation of the progress after the treatment. We performed partial correlations to include the number of sessions in addition to the pretest reading scores as control variables.

First, we looked at crossmodal MMN latency at the initial measurement as it was significantly related to the outcomes of the first part of the reading training [40]. Crossmodal MMN latency at T1 was not significantly related to the combined word and pseudoword reading fluency at the end of the remedial program (*r*_part_ = 0.254, *p* = 0.201). However, crossmodal MMN latency at T1 was significantly related to pseudoword reading fluency at T3 (*r*_part_ = 0.514, *p* = 0.036; Figure 3A), while it was not related to high and low frequency word reading fluency (*r*_part_ = 0.139, *p* = 0.325). It should be noted, however, that the correlation with pseudoword reading fluency was not significant after applying a Bonferroni correction for multiple comparisons for three tests (*α*_bonf_ = 0.016). Thus, this result should interpret with caution. The results could be due to the fact that pseudowords (meaningless non-existent words following phonotactic restrictions of the Dutch language [78]) cannot be visually identified directly via sight word reading, as they are not stored in the visual word form vocabulary, but they can be read by L-SS translation [18]. In this regard, it is possible that the focus of the second part of the training on morphological knowledge [71,100] could orient children from L-SS conversion strategies towards sight word reading [4].

Second, as we previously found that gains in reading fluency are accompanied by decrease of N170 amplitude for words over left parieto-occipital sites [69], we tested if this initial change of N170 during letter-speech sound integration stage of the reading training could predict further gains and outcomes. This analysis yielded a significant result, both for the combined word and pseudoword final fluency outcome of the training (*r*_part_ = −0.642, *p* = 0.009), and for the word (*r*_part_ = −0.666, *p* = 0.006) and pseudoword (*r*_part_ = −0.631, *p* = 0.010) reading fluency separately, indicating that a stronger reduction in N170 amplitudes was associated with better outcome of the reading training. These correlations remained significant after correcting for multiple comparisons (*α*_bonf_ = 0.016). These results extend our previous findings and support the role of visual specialization, i.e., print tuning reflected by N170 changes, in reading fluency. Importantly, this print tuning followed L-SS integration training. Thus, we further investigated relations between letter-speech sound integration and visual specialization.

### 3.2. Relation between Crossmodal MMN and Visual N170

In our data, initial MMN latency, regarded as an indicator of multisensory integration, was significantly correlated with reading fluency. In addition, our visual word study showed that a decrease in N170 amplitudes for words at left occipito-temporal sites from T1 to T2 was correlated with gains in reading fluency. The proposed developmental trajectory of brain networks for reading suggests that successful letter-speech sound integration in multisensory areas facilitates specialization of visual areas for print. In order to examine this, we correlated the MMN latencies at pretest and the post-pretest differences in N170 amplitudes for words (see Figure 4). This correlation was significant (*r* = 0.610, *p* = 0.016) indicating a larger reduction of N170 amplitudes after training in those individuals that showed longer MMN latencies at pretest, the latter reflecting more crossmodal enhancement (see Section 2.1). This result offers partial support for our hypothesis, as it could indicate that dyslexics with less marked integration deficits were more capable of modulating their visual responses after training.

### 3.3. Summary of Results

The findings from our previous ERP studies [38,40,68,69] support the relevance of L-SS integration and visual specialization at the early phases of learning how to read in dyslexic children. In two of those studies, crossmodal MMN latency was related to reading fluency [38] and to reading improvements after training focused on gaining automation in L-SS associations [40]. In the present review, we show that crossmodal MMN latency also correlates with visual tuning for print, as reflected by changes in N170 responses after training, which were previously associated with reading fluency and training outcomes. This is in line with the idea that development of multisensory integration areas facilitates specialization of visual areas for fast letter and word identification [101].

In addition, we evaluated the relation of these ERPs with reading fluency gains in the longer term after completion of the next parts of the program. The results showed a trend (significant at the uncorrected level) associating crossmodal MMN latency to pseudoword reading, but there was no longer a relation between MMN and gains in word reading. A potential explanation of this apparent discrepancy is that as the focus of reading instruction moved to the word level (morphological structure), children adopt different reading strategies, such as sight word reading [4]. That is, they connect orthography directly to meaning [17,18]. At that stage, the importance of a successful L-SS integration is likely to be mainly observed on tasks, such as pseudoword reading, which arguably require a strategy based upon L-SS translation [18]. Visual specialization, on the other hand, would continue to be an important predictor of reading fluency irrespective of the reading strategy, as suggested by the significant correlations between N170 and both word and pseudoword reading in the current paper. This interpretation is in line with previous studies showing that word superiority effect extends to pseudowords (e.g., [102]).

## 4. Enhancing Letter-Speech Sound Binding by Associative, Implicit Learning

We previously discussed the beneficial effects of explicitly addressing L-SS associations in combination with intensive practice, and exposure to gain automation [70]. Our longitudinal ERP studies and current results support a differential and interactive involvement of audiovisual integration and visual specialization in dyslexics’ reading deficits and their response to the L-SS intervention. Another important element we have not yet considered is how audiovisual associations are learned in the first place, before they become automated and temporally integrated. Insights from research on the learning mechanisms allowing for establishing and consolidating L-SS associations could be applied to more efficient training designs and outcome predictors, in particular, given the increasing number of computer-based implicit trainings. General accounts of skill acquisition postulate that the more controlled metacognitive processing in beginners is gradually replaced by more automatic associative processing [103,104]. Accordingly, the L-SS elements are typically taught explicitly and need to be repeated intensively so as to facilitate transition from accurate and controlled processing to associative and automatic processing. Indeed, some studies have suggested that children initially establish L-SS correspondences in an explicit manner but then continue to learn these associations implicitly [105,106]. In this regard, it should be emphasized that learning the alphabetic code (letter knowledge) and learning to associate letters and speech sounds are considered as two different processes. The reading deficits observed in dyslexics seem to be linked to the latter [33,37].

Computer-based trainings based on massive exposure and associative learning are becoming increasingly popular. The aim of these trainings is primarily to improve the persistent reading fluency deficits in dyslexia. Their effectiveness and clinical potential was discussed in a recent review [107]. It should be noted that the authors of this review opted for combining implicit training with initial explicit instruction, which also constitutes a key feature of the treatment discussed in Section 2.2 [70]. Explicit instruction is proposed to act as a bootstrapping mechanism to increase automaticity. This notion is supported by the finding that enhanced reading accuracy after initial stages of specialized treatment, typically explicit, leads to later development of fluency after subsequent treatment stages [108].

In two recent studies, our group examined the diagnostic and prognostic value of a computer-based training of an artificial script [109,110]. The training consisted of a game in which new L-SS associations were implicitly learned, that is, learning the artificial orthography was not the explicit goal of the game. The outcome of the training was used as a dynamic assessment tool. While in a conventional static assessment, the individual’s current knowledge and/or skills are evaluated, dynamic assessment is adaptive and focusing on learning potential and cognitive modifiability. Within this context, trainings typically have a short time frame and contain feedback elements. The scores from the artificial L-SS associations training were able to discriminate between dyslexics and typical readers [110]. Interestingly, these scores could also explain a significant proportion of variance in response to a validated reading intervention program for dyslexia [109]. These studies illustrate the usefulness of the dynamic assessment approach in the context of implicit learning of artificial associations to identify new moderators of response to intervention. In addition, and more closely related to the focus of the present review, a recent neuroimaging study investigated explicit learning of artificial symbol-speech sound correspondences in kindergartners at varying risk of dyslexia [42]. The study demonstrated that a small set of artificial L-SS correspondences could be successfully learned by pre-reading children in less than 30 min. Most importantly, in line with our previous findings, children in that study showed modulation of their brain responses at the posterior systems for reading (temporal and occipito-temporal areas) that were related to their ability to learn the new associations.

The above findings could be interpreted within the multisensory deficit account of dyslexia. It should be noted, however, that besides letter (grapheme) knowledge and phonological skills, there are other cognitive abilities that could contribute to inefficient learning of novel L-SS associations in dyslexics. In fact, several authors proposed a more general learning deficit that is not limited to phonological processing. These authors consider dyslexia as mainly arising from impairments in the procedural learning system for language [111]. In the following sections we present an overview of studies examining dyslexia vis-à-vis higher-order processes that are relevant to associative learning; that is, the detection of statistical regularities and performance monitoring mechanisms (error detection and feedback processing).

## 5. Associative, Implicit Learning and Other Cognitive Processes

In order to efficiently learn and establish L-SS associations, we require a set of cognitive mechanisms that are not limited to the language domain. Some of these mechanisms pertain to bottom-up processes, more involved in implicit learning, while others are related to top-down processes more relevant to explicit learning. As discussed in previous section, both types of learning are proposed to interact during the acquisition of reading skills. Below we review findings from dyslexia studies focusing on different cognitive mechanisms that enable associative learning. The evidence that is currently available points at deficits in general learning and executive functions that may be present in dyslexic readers.

### 5.1. Implicit Learning

Previous studies have suggested that non-linguistic cognitive processes, such as the recognition of statistical properties of stimuli, may be important to implicit associative learning of L-SS correspondences [112,113]. Implicit learning paradigms have widely used the serial reaction time task [114]. In its implicit version, participants are asked to rapidly press a button that corresponds to the location at which a visually presented stimulus appears on screen. Unknown to participants, stimuli are presented in a repeated sequence; the implicit learning of that structure is reflected in faster reaction times across experimental blocks, but subjects are unable to explicitly describe the sequence. Previous studies reported poor performance of dyslexics on implicit serial reaction time tasks [115,116,117]. One of these studies also reported a significant correlation between reading skills and task performance [116]. The findings were interpreted to indicate impaired statistical learning in dyslexic readers. Interestingly, Hedenius and colleagues observed a procedural learning deficit in dyslexic readers and in children with specific language impairment that was related to the consolidation or automatization of the procedural knowledge rather than initial learning [111,118]. This suggests that impairments in dyslexics may also extend to effective long-term learning. Another study found no differences in learning rate, but longer response times in dyslexics, together with a correlation with reading scores that was absent in the control group [119]. Other studies using the implicit serial reaction time paradigm, however, failed to obtain differences in the performance between dyslexics and typical readers [120,121]. These inconsistencies in the literature were discussed in a meta-analysis [122]. The results of that analysis made two important points. First, there is sufficient evidence for considering poor implicit learning abilities as a risk factor for reading problems, and that the interaction between age of participants and methodological characteristics may account for differences between studies. The authors proposed that those differences point to compensatory mechanisms of the declarative memory system. Further, the differences in processing demands for statistical learning in different domains, e.g., audio, visual, tactile, but also audiovisual, were also highlighted in a recent review [123].

Other studies examined implicit learning deficits in dyslexia using an artificial grammar learning (AGL) paradigm [124,125,126,127,128,129], but see [117]. In brief, the AGL paradigm consists of observing sequences of symbols that follow a set of rules (which are not made explicit) and classifying novel sequences as legal or not, according to those rules. Pothos and Kirk [124] used that paradigm with two tasks that presented geometric shapes either serially, to encourage perception of individual items, or embedded, to encourage perception of the stimuli as a whole. The results showed that adult dyslexics performed equally well on both tasks, while controls underperformed dyslexics on the serial task. This was interpreted to reflect interference from explicit attentional mechanisms on the serial task in controls but not in dyslexic readers. More specifically, the authors of the study suggested that difficulties in processing individual elements of stimuli could prevent dyslexics from adopting explicit efforts to “decode” the stimuli in an item-by-item fashion. The study highlights how the interplay between implicit and explicit mechanisms in learning may differ in dyslexics. This was further investigated in a series of studies by Pavlidou and colleagues [125,126,127] in 9–12 year old children. The first study [127] used an AGL task with either implicit instructions (i.e., asking merely to observe training items) or explicit instructions (i.e., asking to memorize as many items as possible). In that study, dyslexics performed worse than controls in both experimental conditions; although they showed some sensitivity to structural regularities of stimuli, they seemed to have difficulties in abstracting this knowledge. This finding was supported by a second study that further controlled for working memory load of the task [126]. However, a study in dyslexic adults found impaired performance only in an implicit AGL task but not when explicit instructions were provided, suggesting an specific deficit in implicit sequential processes [128]. An additional experiment by Pavlidou and colleagues included a transfer task, in which items of a different shape set than the trained ones were tested [125]. The results supported their previous findings, suggesting problems in abstracting higher-order information across stimuli and tasks in the dyslexic group. An important factor in artificial grammar learning studies, especially those in children, is that of task difficulty. A recent study used the same AGL task with an additional manipulation of complexity, as measured by topological entropy [129]. The results showed that decreasing grammar complexity resulted in comparable performance between dyslexics and controls.

Finally, studies used an MMN paradigm (see Section 2) in which the phonotactic probabilities of stimuli are manipulated; that is, the sounds presented differed in their likelihood of occurrence in spoken language [130,131]. The MMN responses in this paradigm are expected to be stronger for stimuli with higher phonotactic probability [132]. Dyslexics showed less sensitivity than typical readers to such enhancement of MMN responses, which was interpreted to reflect a subtle deficit in auditory cortical tuning for phonemic regularities in natural speech [130,131]. Interestingly, some studies in typical readers suggested that lexical knowledge [133] and orthographic information [134] could influence perceptual learning of speech, referred to as phonetic recalibration. The latter study indicated a rapid adjustment of phoneme boundaries induced by the presentation of print-speech pairs [134], while dyslexic readers seemed to be impaired specifically on the print-speech material and not in the more general audiovisual condition, including lip-read [135]. Further studies using a similar paradigm in the context of letter and speech sounds combinations could provide an interesting point of convergence for findings on both phonological and audiovisual learning deficits in dyslexia.

### 5.2. Performance Monitoring

The evaluation of errors and feedback are cognitive control mechanisms instrumental in adapting and optimizing performance in a given task. These mechanisms may, thus, be critical to a successful transition from explicit to implicit and automated, associative learning, when mapping letters and speech sounds. The following paragraphs review a series of dyslexia ERP studies, suggesting deficits in the mechanisms involved in performance monitoring.

#### 5.2.1. Error Processing

The notion of an error detection mechanism is central to theories of executive and cognitive control [136,137]. The efficient detection of erroneous responses is an essential element in various processes important for associative learning, such as performance monitoring and response activation and inhibition. This brain mechanism for error detection is reflected in the error-related negativity (ERN). The ERN has a fronto-central topography and a negative polarity, and is elicited around the first 0–160 ms after erroneous/correct responses. The sources of the ERN seem to be in the anterior cingulate cortex (ACC) in the prefrontal cortex [138]. There are different theories about the functional significance of the ERN: most of them converge on the idea that ERN reflects to some extent a comparison or mismatch between the neural representations of the correct responses and those of the actual response [139,140]. A more recent account includes elements of reinforcement learning and proposes that the ERN may reflect a negative reinforcement signal expressed in the ACC via the mesencephalic dopamine system [141]. Accordingly, this signal is used to optimize performance on a specific task.

There are a few neurophysiological studies reporting deficits in dyslexics in the error detection mechanism, as reflected by the ERN during a lexical decision task [142,143,144,145,146]. One of those studies also reported changes in the ERN responses in adult dyslexics after a reading training [142]. Another study found reduced ERN amplitudes in non-compensated dyslexics compared to compensated dyslexics (i.e., slow readers with relatively well-preserved decoding abilities) [146]. The authors suggested the potential use of ERPs associated with executive functions to evaluate effectiveness of intervention.

#### 5.2.2. Feedback Learning

Related to the ERN, the feedback-related negativity (FRN) is a negative potential elicited between 200 and 300 ms after the presentation of a feedback stimulus, at fronto-central locations. The FRN has been proposed as part of the same reinforcement learning system above [141]. The FRN response has been examined in combination with the P300, a centro-parietal cue-locked positive component thought to reflect attentive processes [147,148]. The FRN-P300 complex has been proposed to index learning ability [149,150]. A recent study examined these responses in dyslexics and controls during the early and late phases of the Wisconsin Card Sorting Task, which relies on both learning and executive functioning [151]. The task performance results showed comparable accuracy in both groups, but slower reaction times in dyslexics compared to controls. The electrophysiological results showed that FRN and P300 amplitudes were smaller during the early phases of the task in dyslexics compared to controls. With regard to the FRN, the initial amplitudes in dyslexics became comparable to those of typical readers during the later phase of the task. Regarding P300, responses in dyslexics decreased during the task in contrast to those of the control group. Based on previous studies, however, both FRN and P300 responses were expected to decrease through task performance due to learning effects and less novelty, respectively [147,148]. Thus, the neurophysiological results in Kraus and Horowitz-Kraus were interpreted by the authors to reflect lower baseline performance that was rapidly adjusted in the dyslexic group.

### 5.3. Conclusions from Associative Learning Studies

The findings reviewed in this section suggest that dyslexics and typical readers may differ in executive functions important for several aspects of associative learning. With regard to implicit learning, we found reports of impairments in bottom-up processes, such as procedural learning, detection of statistical regularities, and phonetic recalibration. In relation to more top-down and explicit processes, there is evidence suggesting that dyslexics may exhibit impairment in error detection and feedback learning mechanisms. These difficulties may be relevant at several stages of fluent reading skills acquisition. For example, they may hinder the establishment and consolidation of L-SS associations, but they may also limit the subsequent use of that information to generate word-specific orthographic representations and to learn exceptions and irregularities (e.g., the self-teaching hypothesis [26]). Furthermore, a central aspect in these mechanisms, especially in those relying on top-down processes, is that of attentional control. This is emphasized in recent attention-based models of associative learning [152,153]. Importantly, there is evidence associating dyslexia with sensory deficits in visual and visuo-spatial attention systems (e.g., [154,155]). The above evidence on executive functioning, associative learning and attentional processes in dyslexia may help in explaining the heterogeneity of the dyslexic population and the high comorbidity with other cognitive disorders, in particular, ADHD [156,157]. Future studies should examine how these mechanisms (attentional and those enabling associative learning) interact to produce the reading deficits observed in dyslexic individuals.

## 6. General Discussion and Future Research

The main goal of this paper was to discuss the combined role of the posterior brain systems for audiovisual integration and visual specialization in dyslexia, and to review findings from associative learning studies that, applied in the context of audiovisual learning, could benefit treatment studies.

We first reviewed and extended the findings of our previous electrophysiological studies. Those studies found two measures that were correlated with reading fluency in dyslexics: the MMN latency in an audiovisual oddball paradigm and the N170 responses to visually-presented words. The continuation of those studies used a pretest-posttest design and a L-SS training, and revealed that those ERP responses also correlated to reading fluency improvements after training [40,69]). In the present review, we included behavioral measurements obtained at a third time point, after children finished the complete remedial program. We extended our previous findings by showing that changes in N170 after the initial part of the treatment also correlated with reading fluency gains program completion, while MMN may still be related to pseudoword reading. We interpreted this pattern of findings to reflect the increasing importance of visual specialization for word recognition as reading fluency develops. This interpretation is related to the notion of a growing reliance on sight word reading with increasing reading expertise [99].

Furthermore, we examined how our two ERP markers were related to each other, and found a significant correlation between initial MMN latencies and changes in N170. This result is supportive of the proposed trajectory of the reading network specialization, namely, that successful L-SS integration facilitates specialization of visual areas. Multisensory areas responsible for letter-speech integration are strongly engaged in the first years of instruction, and interact with the anterior systems that are involved in slow and effortful decoding of new words. Subsequently, as reading expertise increases, the specialization of the ventral system for fast visual print recognition becomes more relevant to fluent word decoding. With regard to the development of the VWFA, strong word selectivity was reported in nine-year-old children [158]. Visual word specialization seems to develop in a relatively short period of time after learning the alphabetic script. A study in six-year-old children showed that word specific VWFA responses could emerged after only a few months of grapheme-phoneme correspondence training [57]. Additionally, similar studies in adults show increased ventral occipito-temporal activations for symbols of artificial scripts after just a few days of training [159,160,161,162]. Importantly, an inverted “U” trajectory has been proposed for VWFA responses [163,164]. Accordingly, activation of visual areas for reading strongly increases at the initial learning stages, but activation declines with reading expertise once print processing becomes more automatic and overlearned. In sum, the mutual interactions and specialization of different functional systems are essential in the development of the reading network. We propose that electrophysiological measures accounting for the functional relation between multisensory integration and visual areas (e.g., functional connectivity studies; [73]) may provide with additional markers of reading (dys)fluency and training outcomes in dyslexia.

There are some limitations in the use of N170 as a marker of visual specialization that should be acknowledged. The use of implicit word recognition tasks may not sufficiently constrain allocation of attentional resources and cognitive strategies during tasks, which can influence early visual components. Indeed, in many of the studies examining occipito-temporal responses to print in dyslexics, the experimental tasks allow for phonological processing of stimuli in addition to orthographic processing (see review in [165]). This has been emphasized in a previous study which intended to restrict as much as possible top-down effects of phonological processing by combining short stimulus duration at high presentation rate with a low-level detection task [166]. The study showed that even restricting high-order processing, dyslexics showed a deficit in print sensitivity of the occipito-temporal areas. The authors interpreted the finding as supporting deficits at bottom-up orthographic processing stages in dyslexics. It is then important to consider task influences when interpreting ERP results based the functionality of VWFA.

The second goal of the present review was to discuss studies investigating cognitive mechanisms involved in associative learning. That type of learning is important in the automation of L-SS associations and it is the focus of a growing number of game-based implicit-learning training programs. There is sufficient evidence to consider deficits in probabilistic learning, error monitoring and feedback learning, as potential contributors to impaired learning of L-SS associations in dyslexia. Such cognitive skills are, thus, relevant to remedial research. The use of dynamic assessment measurements of associative learning and executive functions could provide with additional predictors of treatment outcome and aid in the characterization of dyslexics’ cognitive deficits. In this context, we propose a neurophysiological study using EEG in an experimental task that focuses on learning novel symbol-phoneme associations through feedback after response. The task is based on learning of artificial audiovisual associations as in the study of Aravena and colleagues [34], but in the present case, the learning of stimulus-response associations is explicit, similarly to the probability-learning tasks used in the ERP literature [141]. This paradigm, which is currently being tested in our lab, will allow examining ERPs associated with error/correct responses and positive/negative feedback in the different stages of learning (i.e., at the beginning and at the end of each experimental block). We propose a dynamic balance between feedback-related and error-related brain responses as an additional measure to assess the ability to learn new associations. Moreover, the trial structure, derived from Crone et al. [167], will allow for the evaluation of oscillatory activity (e.g., [168]) and functional connectivity network metrics during tasks [169].

To conclude, learning of audiovisual associations constitutes a crucial step towards reading expertise. The dynamic interplay between L-SS effective integration and visual specialization should be considered to further characterize and predict reading fluency deficits in dyslexics at the different phases of reading instruction, development and trainings. In addition to phonological skills, deficits in other learning mechanisms related to associative learning could hinder the effective consolidation of fluent audiovisual associations, thus having an effect on performance and response to implicit trainings, aimed at gaining fluency. Finally, most research on dyslexia centers around the idea of a primary underlying deficit and the traditional diagnosis excludes other cognitive or neurological dysfunctions. However, the views and evidence emerging in the literature (e.g., attention, executive functions, specific brain areas vs. connectivity) suggest a heterogeneous group characterized by a cluster of deficits, rather than a homogenous group with a single underlying deficit (e.g., [170]).

## Figures and Tables

**Figure 1 brainsci-07-00010-f001:**
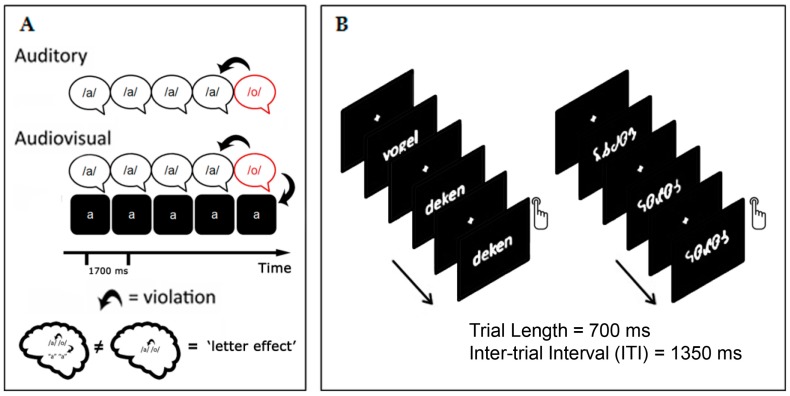
Crossmodal MMN paradigm (**A**); and visual word recognition paradigm consisting of a one-back task with words and symbol strings (**B**).

**Figure 2 brainsci-07-00010-f002:**
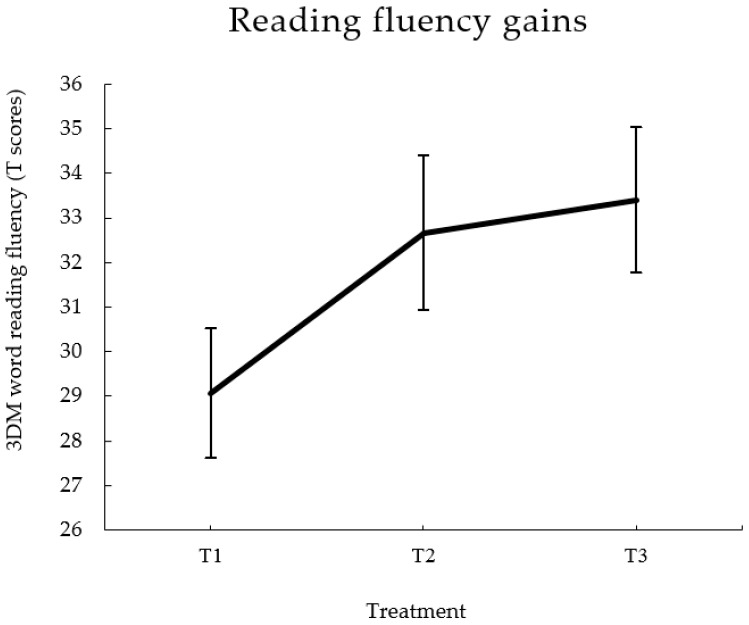
Reading fluency scores (3DM total T scores; words and pseudowords combined) before starting training (T1), after the letter-speech sound training (T2), and after completion of the full remedial program (T3).

**Figure 3 brainsci-07-00010-f003:**
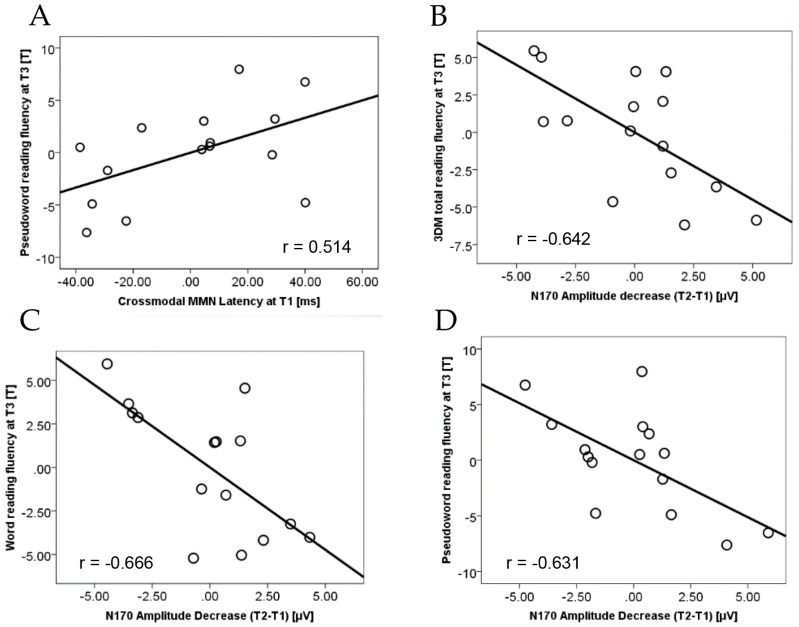
(**A**) Crossmodal MMN latency at pretest (T1) was significantly related to standardized pseudoword reading fluency scores at the end of the training (T3) when controlling for pseudoword reading fluency at T1 and number of sessions. (**B**) Decrease in N170 amplitude from pretest (T1) to the end of the first part of the letter-speech sound training (T2) was significantly related to the combined word and pseudoword, and (**C**) word and (**D**) pseudoword reading fluency scores at the end of the training (T3) when controlling for the appropriate fluency scores at T1 and number of sessions.

**Figure 4 brainsci-07-00010-f004:**
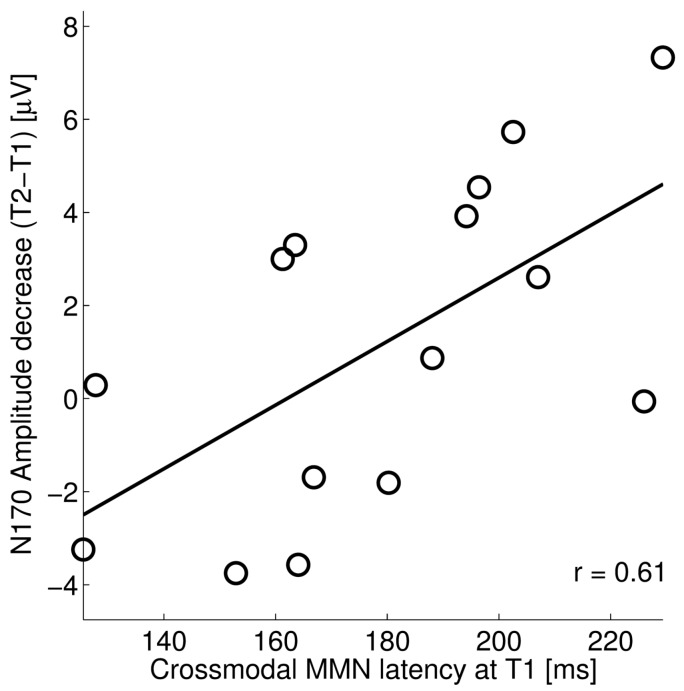
Crossmodal MMN latency at pretest (T1) was related to the N170 amplitude decrease from pretest (T1) to the end of the first part of the training, which was aimed at letter-speech sound integration (T2). A change towards positive values along the y-axis refers to a decrease in the N170 amplitude.

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
