# Peer review of "Contributions of Letter-Speech Sound Learning and Visual Print Tuning to Reading Improvement: Evidence from Brain Potential and Dyslexia Training Studies"

_brainsci, 2017, doi:10.3390/brainsci7010010_

Round 1

Reviewer 1 Report

The manuscript provides an overview of neurophysiological research on dyslexia and remedial training. It is comprehensive and ties the current research together well.

Major comments:

Section 2, line 154. The LN is introduced but never described. It’s referred to in various parts of the text. It would be helpful to give a description of what this component is thought to reflect, in order to understand what LN modulation means.

Line 206/207. “This is supported by two recent studies on L-SS integration in dyslexic children.” A brief statement about how it is supported would be helpful.

Section 3. Were all tests corrected for multiple comparisons?

Line 379 and 384. I’m unclear on what “dynamic assessment” means. An explicit definition would help to understand what “dynamic assessment tool” and “dynamic assessment approach” mean.

Section 5. I would like to see some discussion of the discrepancy between the initial description of dyslexia on lines 46/47, and the fact that there do seem to be more general learning deficits.

Section 5.2. Is there any evidence to support performance monitoring being impaired in dyslexia?

Minor comments:

Line 77 “While familiar words are read without involvement of phonology and unfamiliar words and pseudowords would require orthographic-phonological decoding (e.g. [25,26]).”

            Sentence structure is incorrect. Should probably say While familiar words are read without involvement of phonology, and unfamiliar words and pseudowords would require orthographic-phonological decoding (e.g. [25,26]).”

Line 130 “pertaining relating” should be one or the other

Line 179 there should not be a comma after “dyslexics”

Line 362 “indeed” is used twice in the sentence.

Author Response

We would like to thank the reviewers for a prompt review and the useful comments on our manuscript. We have addressed most of their points and we believe that the changes made improved our manuscript. Please see below our responses to the comments raised by the reviewers and the changes made to the manuscript (in track changes).

  In addition, we noticed that the sign of Y-axis of figure 4 was reversed, displaying the T1-T2 instead of T2-T1 difference in N170 amplitudes. We have now corrected this and added the following line in the footnote for clarification: “A change towards positive values along the y-axis refers to a decrease in N170 amplitude.” Note that the sign of the correlation changed but the interpretation in the previous version of the manuscript was correct. 

 Major comments:

Section 2, line 154. The LN is introduced but never described. It’s referred to in various parts of the text. It would be helpful to give a description of what this component is thought to reflect, in order to understand what LN modulation means.

 We have included that information in the paragraph; “The LN is a later negativity within a broader time range that is usually found around 300-700 ms after the onset of the deviant stimulus. (…). The LN is hypothesized to reflect the more cognitive and/or attentional aspects of this integration, although its functional role is still unclear”

 Line 206/207. “This is supported by two recent studies on L-SS integration in dyslexic children.” A brief statement about how it is supported would be helpful.

 We have added a reference to findings reported in these studies: “This is supported by two recent ERP studies on L-SS integration in dyslexic children that observed abnormal responses suggestive of greater cognitive effort rather than automaticity [91] and a delay in using orthographic and phonological information during response selection [38].

  Section 3. Were all tests corrected for multiple comparisons?

 We now applied Bonferroni correction for our correlational results. We used the Bonferroni correction considering 3 comparisons (combined reading fluency scores, high and low frequency words, and pseudowords) for each of the ERP markers. While the correlations of N170 with reading scores remained significant, the correlation of MMN latency with pseudowords did not reach significance (αbonf=.016). As we still consider this result interesting, we retained the interpretation of the relation but downplayed its weight considering the significance level.

 Line 379 and 384. I’m unclear on what “dynamic assessment” means. An explicit definition would help to understand what “dynamic assessment tool” and “dynamic assessment approach” mean.  

 We have included a clarifying line in that paragraph: “The outcome of the training was used as a dynamic assessment tool. While in conventional static assessment the individual’s current knowledge and/or skills are evaluated, dynamic assessment is adaptive and focusing on learning potential and cognitive modifiability.”

  Section 5. I would like to see some discussion of the discrepancy between the initial description of dyslexia on lines 46/47, and the fact that there do seem to be more general learning deficits.

 We have added the following concluding lines:

“Finally, most research on dyslexia centers around the idea of a primary underlying deficit and the traditional diagnosis excludes other cognitive or neurological dysfunctions. However, current views and evidence emerging in the literature (e.g., attention, executive functions, specific brain areas vs. connectivity) suggest a heterogeneous group characterized by a cluster of deficits, rather than a homogenous group with a single underlying deficit ( e.g., [169]).”

 Section 5.2. Is there any evidence to support performance monitoring being impaired in dyslexia?

 We consider the findings presented in section 5.2. from ERP studies of dyslexia on error detection and feedback related components as evidence that there may be deficits in dyslexia in performance monitoring. To make that point more explicit we added the following sentence in the introduction of point 5.2. “The following paragraphs review a series of dyslexia ERP studies suggesting deficits in the mechanisms involved in performance monitoring.”

 Minor comments:

 Line 77 “While familiar words are read without involvement of phonology and unfamiliar words and pseudowords would require orthographic-phonological decoding (e.g. [25,26]).”

            Sentence structure is incorrect. Should probably say “While familiar words are read without involvement of phonology, and unfamiliar words and pseudowords would require orthographic-phonological decoding (e.g. [25,26]).”

 That sentence has been corrected

 Line 130 “pertaining relating” should be one or the other

  This sentence has been corrected, ‘relating’ has been removed.

 Line 179 there should not be a comma after “dyslexics”.

 This sentence has been corrected

Line 362 “indeed” is used twice in the sentence.

 This sentence has been corrected

Reviewer 2 Report

This review paper on predictors and precursors of reading acquisition was very interesting and relevant for the field. The manuscript is written clearly and structured in such a way that it is easy to follow. There are some minor issues that would need revising, detailed below.

In general, the review focuses mostly on the work conducted by the authors. This is not a bad thing in itself, but left the impression that the review did not thoroughly compare the findings to research done by other groups. There are some references to for example work by Brem et al., but then there are missing references to e.g., Nigro et al., 2015 Annals of dyslexia, Kahta et al., 2016 Annals of dyslexia, Katan et al., 2016 Annals of dyslexia, Vakil et al., 2015 Journal of learning disabilities. I would suggest the authors either justify the limited scope of the review or integrate as much of the previous studies into it as possible.

Another potentially important point is the description of the additional analyses on the training studies. It was not clear if the comparison to the T3 time is reported in more detail elsewhere. If it is not, then more details should be described in the current paper: What was the severity of the dyslexia (reading test scores) and age of the children. Also it was not clear what kind of training material was used after T2, were the same words used in the remediation as in the reading tests? This would explain why only the reading but not pseudoword reading scores got better. How strongly the different reading scores correlated at each time point? This would be important for interpretation of the ERP correlations to reading (e.g., if the reading scores show high correlation between time points then it is not surprising that the ERP score that correlated with reading at T2 would also correlate with reading at T3).

Line566: the inverted U trajectory is seen for the N170 amplitudes and not visual expertise (it is likely that the expertise shows an increasing trend through development even when the N170 amplitude does not).

There are some spelling errors:

Line 199: disfluent -> dysfluent

Line 332: N1 -> N170

Line 519: functions important several aspects -> functions important for several...

Line 587: involve -> involved

Author Response

We would like to thank the reviewers for a prompt review and the useful comments on our manuscript. We have addressed most of their points and we believe that the changes made improved our manuscript. Please see below our responses to the comments raised by the reviewers and the changes made to the manuscript (in track changes).

 In addition, we noticed that the sign of Y-axis of figure 4 was reversed, displaying the T1-T2 instead of T2-T1 difference in N170 amplitudes. We have now corrected this and added the following line in the footnote for clarification: “A change towards positive values along the y-axis refers to a decrease in N170 amplitude.” Note that the sign of the correlation changed but the interpretation in the previous version of the manuscript was correct. 

Reviewer 2

Comments and Suggestions for Authors

This review paper on predictors and precursors of reading acquisition was very interesting and relevant for the field. The manuscript is written clearly and structured in such a way that it is easy to follow. There are some minor issues that would need revising, detailed below.

In general, the review focuses mostly on the work conducted by the authors. This is not a bad thing in itself, but left the impression that the review did not thoroughly compare the findings to research done by other groups. There are some references to for example work by Brem et al., but then there are missing references to e.g., Nigro et al., 2015 Annals of dyslexia, Kahta et al., 2016 Annals of dyslexia, Katan et al., 2016 Annals of dyslexia, Vakil et al., 2015 Journal of learning disabilities. I would suggest the authors either justify the limited scope of the review or integrate as much of the previous studies into it as possible.

We appreciate the new references and have included them in the text of section 5.1.  In the previous parts of the manuscript (e.g. section 2 and 3), our main focus is on the literature using EEG/MEG and similar paradigm to our own and reporting findings of multisensory integration and N170. The literature review in those sections primarily aims at providing the background for our previous and current analysis, while in section 5 we tried to provide a broader literature review.

Another potentially important point is the description of the additional analyses on the training studies. It was not clear if the comparison to the T3 time is reported in more detail elsewhere. If it is not, then more details should be described in the current paper: What was the severity of the dyslexia (reading test scores) and age of the children. Also it was not clear what kind of training material was used after T2, were the same words used in the remediation as in the reading tests? This would explain why only the reading but not pseudoword reading scores got better. How strongly the different reading scores correlated at each time point? This would be important for interpretation of the ERP correlations to reading (e.g., if the reading scores show high correlation between time points then it is not surprising that the ERP score that correlated with reading at T2 would also correlate with reading at T3).

The T3 comparison in the current sample was not reported previously. However, the T2-T3 training corresponds to the module 3 of the training evaluated in Tijms , 2004 (the reader is referred to that paper for further details). We have now included the age of the subsample analyzed and a supplementary table containing their reading scores at T1,T2 and T3, as well as the t-tests for each of those tests for T2-T1 and T3-T2. Because the focus of the present work is  on correlations with ERP markers, we prefer to include the table as supplementary material.

Regarding the material used in the training, the words of the tests are not systematically incorporated in the training material. An incidental exposure during the training to a word appearing in the tests would be comparable to the frequency of exposure in text at school. Note that the current training is not a word-trainer and focuses on sub-word level knowledge needed for reading and spelling in Dutch. This implies that the training consists of two corpora of words, one for the instruction part of the training and the other, consisting of different words, for the practice part of the training. It also implies that repeated exposure is not at the word-level (i.e. the same word is not practiced over and over again), but at sub-word level (e.g. repeated practice of a corpus of words with CVC structure including a short vowel, or words with a certain type of bound morpheme). Both corpora do not match with the words used in the tests for evaluating training progress.

Finally, it is indeed the case that reading scores are strongly correlated between time points. These correlations are stronger between T1 and T2, ranging from .75 to .85. We believe that our correlational analysis with an additional time point is still interesting, as a) it can show whether these ERP markers would still be valid predictors of reading gains in a broader time window than in our previous studies and b) it illustrates how both markers (N170 and MMN) preform as predictors (in the same sample); this point is further developed in our correlation between N170 and MMN. 

Line566: the inverted U trajectory is seen for the N170 amplitudes and not visual expertise (it is likely that the expertise shows an increasing trend through development even when the N170 amplitude does not).

Indeed, that sentence may be confusing, now it reads: “Importantly, an inverted “U” trajectory has been proposed for VWFA responses [161,162].” Note that the following sentence further elaborates: “Accordingly, activation of visual areas for reading strongly increases at the initial learning stages, but activation declines with reading expertise once print processing becomes more automatic and overlearned.”

There are some spelling errors:

Line 199: disfluent -> dysfluent

Line 332: N1 -> N170

Line 519: functions important several aspects -> functions important for several...

Line 587: involve -> involved

These spelling errors have been corrected

Round 2

Reviewer 1 Report

The authors have addressed all of my concerns.